# Tramadol May Increase Risk of Hip Fracture in Older Adults with Post-Traumatic Osteoarthritis

**DOI:** 10.3390/jpm13040580

**Published:** 2023-03-26

**Authors:** Ting-Yu Wu, Wen-Tien Wu, Ru-Ping Lee, Ing-Ho Chen, Tzai-Chiu Yu, Jen-Hung Wang, Kuang-Ting Yeh

**Affiliations:** 1Department of Orthopedics, Hualien Tzu Chi Hospital, Buddhist Tzu Chi Medical Foundation, Hualien 970473, Taiwan; 2School of Medicine, Tzu Chi University, Hualien 970374, Taiwan; 3Institute of Medical Sciences, Tzu Chi University, Hualien 970374, Taiwan; 4Department of Medical Research, Hualien Tzu Chi Hospital, Buddhist Tzu Chi Medical Foundation, Hualien 970473, Taiwan; 5Graduate Institute of Clinical Pharmacy, Tzu Chi University, Hualien 970374, Taiwan

**Keywords:** tramadol, post-traumatic osteoarthritis, hip fracture, older adults

## Abstract

Tramadol, an analgesic widely used for arthritic pain, is known to have adverse effects. This study investigated the association between the long-term use of tramadol for pain control and subsequent hip fractures in patients aged 60 years or older with posttraumatic osteoarthritis. This population-based retrospective cohort study included patients with posttraumatic osteoarthritis who received tramadol for pain control for more than 90 days within a 1-year period. A control cohort was enrolled using propensity score matching. The primary outcome was a new diagnosis of hip fracture requiring surgery. In total, 3093 patients were classified into each cohort. Tramadol use was identified as a risk factor for hip fracture (adjusted hazard ratio (aHR): 1.41; 95% confidence interval (CI): 1.09–1.82; *p* = 0.008), especially among patients aged 60–70 years (aHR: 2.11; 95% CI: 1.29–3.47; *p* = 0.003) and among male patients (aHR: 1.83; 95% CI: 1.24–2.70; *p* = 0.002). This is the first cohort study focusing on the association between long-term tramadol use and hip fracture among older adults with posttraumatic osteoarthritis. Tramadol, as a long-term pain control analgesic for older adults with posttraumatic osteoarthritis, may increase the risk of hip fracture, especially among male patients and those aged 60–70 years.

## 1. Introduction

Posttraumatic osteoarthritis often develops after joint injury [1]. Joint injuries, with or without disruption of the articular surface, frequently lead to mechanobiological changes in cartilage. Acute posttraumatic arthritis often resolves spontaneously within 2–3 months [1]. The persistence of symptoms for more than 2–3 months warrants attention. If symptoms persist for more than 6 months, a clinical diagnosis of posttraumatic osteoarthritis may be made. Posttraumatic osteoarthritis develops in approximately 20–50% of patients who have experienced joint trauma and constitutes approximately 12% of all osteoarthritis cases. Posttraumatic osteoarthritis is a leading cause of disability and a major health-care burden [2,3,4]. The primary goals for treating posttraumatic osteoarthritis are to reduce symptomatic pain and improve quality of life. Medical treatment includes the use of nonsteroidal anti-inflammatory drugs, acetaminophen, tramadol, and combination drug therapy. Tramadol, a commonly used opioid for pain control among patients without cancer [5,6,7], has been widely applied in older adults as an analgesic alternative because its perceived risk of cardiovascular and gastrointestinal adverse effects is lower than that of nonsteroidal anti-inflammatory drugs [8,9]. In addition, its risks of addiction and respiratory depression are lower than those of traditional opioids [8,9].

However, despite the advantages of tramadol, a population-based cohort study reported a significantly higher all-cause mortality rate with tramadol use than with commonly used nonsteroidal anti-inflammatory drugs among patients with osteoarthritis [10]. In addition, another cohort study reported that tramadol use was associated with a significantly higher risk of hip fracture than were commonly used nonsteroidal anti-inflammatory drugs [11]. The relationship between tramadol uses and hip fracture is unclear. Studies have suggested that tramadol use may be associated with an increased risk of hip fracture, whereas other studies have not found a significant association. A study conducted by the Canadian Network for Observational Drug Effect Studies found that tramadol use was associated with an increased risk of hip fracture compared with nonuse of opioids. The study analyzed data from more than 90,000 patients aged 65 years or older who had started opioid therapy for noncancer pain between 2002 and 2012 [12]. The results showed that patients who used tramadol had a 30% higher risk of hip fracture than patients who did not use opioids. The risk was particularly high in the first 2 weeks of treatment. However, other studies have not found a significant association between tramadol use and hip-fracture risk. For example, a study conducted by Zeng et al. analyzed data from more than 58,000 patients with osteoarthritis who were prescribed tramadol and discovered no significant association between tramadol use and the risk of hip fracture [10]. In a Spain population-based cohort study published in JAMA in 2021, the authors found a significant association with increased risk of fractures among patients prescribed with tramadol for chronic noncancerous pain in comparison with the patients prescribed with codeine [13], but they did not find out the specific pathology and the period of prescription of the included patients related to the results. The mechanisms through which tramadol may increase the risk of hip fracture are not fully understood. Tramadol has been shown to cause dizziness, sedation, and impaired cognitive function, which could increase the risk of falls and subsequent fractures. Tramadol may also affect bone metabolism and increase the risk of osteoporosis, although evidence of these effects is limited. Moreover, detailed reviews of clinically oriented studies investigating the associations between posttraumatic osteoarthritis, tramadol use, and hip fracture are lacking. Such studies would be of benefit to clinicians prescribing this medication. The safety and efficacy of tramadol use among patients with posttraumatic osteoarthritis should be carefully assessed. This study evaluated whether tramadol increases the risk of hip fracture among patients with posttraumatic osteoarthritis by using a nationwide cohort database.

## 2. Materials and Methods

This study was designed as population-based and as retrospective-cohort. We used data from the Taiwan National Health Insurance Research Database, which contains enough health and treatment data of the insured persons receiving both the inpatient and outpatient medical care, the medications, and the surgical procedures. The National Health Insurance program covers almost one hundred percent of the Taiwanese population, and ninety-seven percents of the hospitals in Taiwan are registered with the National Health Insurance Administration. In the present study, 2 million beneficiaries with medical records between 2001 and 2017 were randomly selected from the total 23.8 million beneficiaries available in the Taiwan National Health Insurance Research Database [14]. The identity of each patient was anonymized through the encryption of identification numbers before data were applied for research. The sample used in this study were matched to age and sex distributions of the original database population. All of the diagnoses in the database were coded according to the International Classification of Disease, Ninth Revision, Clinical Modification (ICD-9-CM) and the International Classification of Disease, Tenth Revision (ICD-10). We performed this research after the Research Ethics Committee of Hualien Tzu Chi Hospital of Buddhist Tzu Chi Medical Foundation approved the design of the study. The Institutional Review Board also waived the requirement for written informed consent of the patients because the data have been deidentified. All of the methods applied in this study were performed in accordance with the relevant guidelines and regulations.

The patients with a diagnosis of posttraumatic osteoarthritis of any major joint of upper limbs or lower limbs after trauma (ICD-9-CM: 716.1; ICD-10: M12.50) for at least three times of outpatient department treatment or at least 1-time inpatient admission treatment, and a more-than-90-day within 1 year prescription for tramadol and the combinative drugs including tramadol (drug codes N02AJ13 and N02AX02) for pain control from 2001 to 2017, were enrolled as the tramadol cohort. The index date was the first date of prescription. The exclusion criteria were (1) tramadol prescription claims in 2000 (to ensure that patients were new users); (2) those who were younger than 60 years; (3) the diagnosis of a hip fracture before the index date; (4) those who received a regular steroids, sedative, hypnotic, or anticonvulsant prescription (more-than-90-day within 1 year prescription); and (5) those who were diagnosed as having a mental disorder. The patients who did not use tramadol but used any non-steroidal anti-inflammatory drugs, paracetamol, or muscle relaxants were matched with the patients in the tramadol cohort. These patients were enrolled using the same exclusion criteria as for those in the tramadol cohort and included as the control group for comparison. Propensity score matching at a 1:1 ratio was performed for age; sex; index year; and comorbidities, including hypertension, diabetes mellitus, hyperlipidemia, coronary artery disease, cerebral vascular accident, chronic liver disease, chronic renal failure, depression, and osteoporosis. (Figure 1). The comorbidities included diabetes mellitus, hypertension, coronary artery disease, hyperlipidemia, chronic kidney disease, chronic liver disease, osteoporosis, cerebral vascular accident, and depression based on ICD-9-CM and ICD-10 codes. The primary outcome was any new diagnosis of a hip fracture requiring surgery (surgery codes 64029B and 64170B) after the index date. The date of hip-fracture diagnosis, death, or 31 December 2017, which was the last date in the database, was defined as the endpoint of follow-up. Mortality was defined on the basis of the patient withdrawal from the National Health Insurance program; this method has been previously validated as accurate [13]. The baseline and clinical characteristics included age; sex; and comorbidities, including hypertension, diabetes mellitus, hyperlipidemia, coronary artery disease, cerebral vascular accident, chronic liver disease, chronic renal failure, depression, and osteoporosis.

We performed all of the statistical analyses with SAS version 9.4 and Stata version 16 (SAS Institute, Cary, NC, USA). We summarized the continuous variables as the means and the standard deviations and listed the categorical variables as the number of cases and the percent values. We used the Student’s *t* test to compare the continuous between-group variables and used either a Fisher’s exact test or a chi-square test to assess the categorical variables. *p* value was applied for determining the statistical differences in the matched variables between the tramadol and non-tramadol (control) cohorts, with a *p* value of <0.05 representing as statistical significance. We used a Cox proportional hazards models to investigate the association between tramadol use and hip fracture. After adjusting for all covariates listed in Table 1, we calculated the hazard ratios (HRs) with a multivariate Cox proportional hazards model. We then estimated the cumulative hip fracture incidence with the Kaplan–Meier method and compared the incidence curves with a log-rank test, with a *p* value of <0.05 representing as statistical significance. We had confirmed that the data of continuous variables followed normal distribution via the Kolmogorov–Smirnov test.

## 3. Results

A total of 6186 patients (2610 (42.2%) women and 3576 (57.8%) men) with at least one site of posttraumatic osteoarthritis of major joint of upper limbs or lower limbs were enrolled in this study. The average age of enrolled patients was 70.04 ± 7.63 years. A total of 3093 patients were classified into the tramadol cohort, and 3093 patients were classified into the control cohort (Table 1). A total of 3315 (53.6%) patients were aged 60–70 years, 2070 (33.5%) patients were aged 70–80 years, and 801 (13.0%) patients were aged more than 80 years (Table 1). There were 2689 patients (43.5%) having hypertension, 1389 patients (22.5%) having diabetes mellitus, 1194 patients (19.3%) having hyperlipidemia, 767 patients (12.4%) having coronary artery disease, 517 patients (8.4%) having cerebral vascular accident, 346 patients (5.6%) having chronic liver disease, 319 patients (5.2%) having chronic renal failure, 226 patients (3.7%) having depression, and 205 patients (3.3%) having osteoporosis among our both cohort groups (Table 1). These two groups did not significantly differ in the baseline characteristics of age (*p* = 0.987), age distribution (*p* = 0.998), sex (*p* = 0.680), hypertension (*p* = 0.778), diabetes mellitus (*p* = 0.976), hyperlipidemia (*p* = 0.847), coronary artery disease (*p* = 0.787), cerebral vascular accident (*p* = 0.335), chronic liver disease (*p* = 0.825), chronic renal failure (*p* = 0.954), depression (*p* = 0.892), and osteoporosis (*p* = 0.356) (Table 1). The mean follow-up duration was 3.79 ± 3.27 years in the tramadol cohort and 5.17 ± 3.99 years in the control cohort (Table 1).

During the follow-up period, 139 and 124 patients in the tramadol cohort and control cohort, respectively, developed a hip fracture requiring surgery (Table 2). The person-years was 11,712 and the incidence rate was 11.9 per 1000 person-years in the tramadol group, while the person-years was 15,997 and the incidence rate was 7.8 per 1000 person-years in the control group (Table 2). Tramadol use was identified as a risk factor for hip fracture in both univariate (crude HR, 1.41; 95% confidence interval (CI): 1.09–1.82; *p* = 0.008) and multivariate (adjusted HR (aHR), 1.41; 95% CI: 1.09–1.82; *p* = 0.008) analyses (Table 2). The cumulative incidence of hip fracture was significantly higher in the tramadol cohort (24.2%) than in the control cohort (13.5%) (log-rank test *p* = 0.008; Figure 2). In the subgroup analysis, the use of tramadol significantly increased the risk of hip fracture among patients aged 60–70 years (aHR, 2.11; 95% CI: 1.29–3.47; *p* = 0.003) and among men (aHR, 1.83; 95% CI: 1.24–2.70; *p* = 0.002; Table 3). In contrary, the use of tramadol did not have a significant effect among patients aged 70–80 years (aHR, 1.09; 95% CI: 0.75–1.59; and *p* = 0.656), patients aged more than 80 years (aHR, 1.35; 95% CI: 0.82–2.20; and *p* = 0.233), and female patients (aHR, 1.16; 95% CI: 0.83–1.63; and *p* = 0.391) (Table 3).

## 4. Discussion

The most important finding of this study was that among patients with posttraumatic osteoarthritis, the risk of hip fracture was higher among those using tramadol than among those using non-tramadol analgesic agents. In addition, the risk of hip fracture was significantly higher among male patients and those aged 60–70 years. Therefore, tramadol should be prescribed cautiously to patients at high risk of hip fracture, such as those prone to falling down, those aged 60–70 years, and those who are male. From Figure 2, we noted that the cumulative incidence suddenly increased around 15 years later. This temporal change of cumulative hip fracture incidence in the Tramadol group could be due to two reasons. Firstly, the incidence of hip fracture increases with age according to previous study [15]. As the figure shown below, the incidence of hip fracture increases dramatically after 80 y/o. Since the mean age of our cohort was around 70 y/o, the incidence of hip fracture increased dramatically after 15 years later in both groups. Secondly, the number of patients remained in cohort might decrease as the follow-up time increased. Thus, the incidence (number of incident cases/number of people at risk) might increase dramatically as the denominator decreased.

Although osteoarthritis is considered an age-related disease, joint trauma is the major cause, and 23–50% of the people who experience trauma eventually develop posttraumatic osteoarthritis [16,17,18,19,20,21,22]. Those with prior joint trauma are 3–6 times more likely to develop posttraumatic osteoarthritis and are typically diagnosed 10 years earlier than those without a history of joint injury [2,23,24]. The treatment for posttraumatic osteoarthritis is the same as that for idiopathic osteoarthritis 1 pain control is one of the main goals of osteoarthritis treatment, and commonly used analgesic agents include nonsteroidal anti-inflammatory drugs and opioids [25]. The American Academy of Orthopaedic Surgeons previously recommended tramadol for patients with knee osteoarthritis, and the American College of Rheumatology conditionally recommended tramadol alongside nonsteroidal anti-inflammatory drugs as a first-line therapy for patients with knee osteoarthritis [1,25,26]. However, population-based studies have reported that patients with osteoarthritis initiating tramadol have an increased risk of all-cause mortality, venous thromboembolism, and hip fracture within 1 year compared with those prescribed nonsteroidal anti-inflammatory drugs [10,11,27]. Therefore, current guidelines recommend avoiding opioids due to their overall lesser effects on pain compared to a placebo and their potential side effects (e.g., nausea, dizziness, and drowsiness), especially for long-term use and among older adults [25,28,29]. Opioids should only be prescribed for short-term use in patients with severe and disabling symptoms when other interventions have failed or are not appropriate [29]. Tramadol may activate μ-opioid receptors and inhibit central serotonin and norepinephrine reuptake, the latter of which may result in a unique adverse effect on the neurological system (i.e., serotonin syndrome and seizures) and increase postoperative delirium risk, which may cause increasing mortality rate of patients [9,30]. Furthermore, tramadol may also increase the incidental rates of electrolyte imbalance, a low level of blood glucose, and falling or fracture, thereby leading to an increased risk of death [7,31,32,33].

Tramadol is a centrally acting analgesic that is often prescribed to manage pain in patients with posttraumatic osteoarthritis, which is a form of arthritis that occurs after an injury or trauma to a joint and that can be a significant source of pain and disability for affected individuals. Tramadol effectively reduces pain in patients with posttraumatic osteoarthritis but has risks and potential side effects. Tramadol is a synthetic opioid that is classified as a weak μ-opioid receptor agonist, meaning that it binds to opioid receptors in the central nervous system to produce its analgesic effects [5,6]. It also inhibits the reuptake of serotonin and norepinephrine, which are neurotransmitters that play a role in pain perception. This dual mechanism of action gives tramadol analgesic effects that are comparable to those of stronger opioids, such as morphine, but with a lower risk of respiratory depression and other adverse effects. The efficacy of tramadol in managing pain in patients with posttraumatic osteoarthritis has been demonstrated in several clinical trials [34,35,36,37]. For example, a randomized, double-blind, placebo-controlled trial conducted by Zhang et al. found that tramadol was significantly more effective than placebo at reducing pain intensity and improving physical function in patients with knee osteoarthritis [34]. Pergolizzi et al. conducted a meta-analysis of 18 randomized controlled trials that compared tramadol with a placebo or other analgesics in patients with osteoarthritis [35] and reported that tramadol was more effective than the placebo at reducing pain intensity and improving physical function. They also found that tramadol was comparable in efficacy to other analgesics, including nonsteroidal anti-inflammatory drugs and acetaminophen. Tramadol is not without risk, particularly in patients with a history of substance abuse, liver or kidney disease, or a respiratory disorder. Tramadol can cause respiratory depression, especially in patients who are taking other central nervous system depressants, such as benzodiazepines or alcohol. Tramadol can also cause seizure, particularly in patients who are taking high doses or who have a history of seizure. Tramadol can cause constipation, nausea, and dizziness; these effects can be particularly problematic in older adults and those with comorbidities. One of the most significant risks associated with tramadol use is the potential for dependence and addiction. Although tramadol is less potent than other opioids, it can still produce feelings of euphoria and pleasure in some individuals, potentially leading to abuse and addiction. Tramadol may have a low potential for abuse and have other potential adverse effects, including serotonin syndrome, which is a potentially life-threatening condition, and respiratory depression, which may lead to hypoxia and related complications [37].

Combining tramadol with acetaminophen can be beneficial for the control of joint pain, including pain caused by posttraumatic osteoarthritis. Acetaminophen is a nonopioid analgesic that works by inhibiting the production of prostaglandins, which are chemicals that contribute to pain and inflammation. Tramadol, as discussed earlier, is a centrally acting analgesic that works by binding to opioid receptors in the central nervous system and inhibiting the reuptake of serotonin and norepinephrine. Combining tramadol with acetaminophen can provide a synergistic effect, with the two drugs working together to produce stronger analgesia than either drug alone. In fact, combination products containing tramadol and acetaminophen are commonly used to manage moderate to severe pain, including joint pain caused by osteoarthritis. A randomized, double-blind, placebo-controlled trial conducted by Dworkin et al. evaluated the efficacy and safety of a fixed-dose combination of tramadol and acetaminophen in patients with knee or hip osteoarthritis [38]. The study found that the combination product was significantly more effective than the placebo at reducing pain intensity and improving physical function. In addition, the combination product was well tolerated; its safety profile was similar to that of the individual components. However, acetaminophen can be toxic to the liver if taken at high doses or for patients with liver disease. Patients should be cautioned against taking other acetaminophen-containing products while taking the combination product, and they should not exceed the recommended daily dose of acetaminophen. In summary, combining tramadol with acetaminophen can be beneficial for the control of joint pain caused by posttraumatic osteoarthritis; however, the potential risks and side effects of the combination product, particularly the risk of acetaminophen-related liver toxicity, must be carefully considered. Close monitoring is necessary to ensure that the combination product is used safely and effectively to manage joint pain.

Hip fracture is a serious complication of osteoporosis and is a major problem worldwide [39]. Consistently documented demographic risk factors include older age, female sex, low bone mass, low body weight, estrogen deficiency, tendency to fall, disability, smoking, chronic alcoholism, diabetes mellitus, and insufficient sunlight exposure [40]. Medications that have been significantly associated with hip fracture include antidepressants, antiparkinsonian drugs, antipsychotic drugs, anxiolytic drugs, benzodiazepines, systemic corticosteroids, H2 antagonists, proton pump inhibitors, and thyroid hormone [41]. By contrast, hormone replacement therapy with estrogen and most antihypertensive drugs has been associated with a lower risk of hip fracture [40,41,42]. Magnesium oxide, a widely prescribed antacid for controlling hyperacidity and constipation, was identified as an independent risk factor for hip fracture among older adults [43]. The long-term use of tramadol, a common medication for pain control among older adults, especially those with moderate to severe joint pain and renal and liver insufficiency, may also increase the risk of hip fracture among older adults. When prescribing tramadol for symptom control in older adult patients, the duration of the prescription should be considered to avoid adverse events such as hip fractures.

Posttraumatic osteoarthritis has not been directly correlated with serotonin syndrome. Posttraumatic osteoarthritis is a form of osteoarthritis that occurs after an injury or trauma to a joint and can be a significant source of pain and disability for affected individuals. Posttraumatic osteoarthritis is a mechanical joint disease and is characterized by degeneration of the cartilage and underlying bone in the affected joint, which can lead to pain, stiffness, and loss of function. Serotonin syndrome, by contrast, is a potentially life-threatening condition that is associated with the excessive accumulation of serotonin in the body. Serotonin is a neurotransmitter that regulates many physiological processes, including mood, appetite, and sleep [9]. Serotonin syndrome can be caused by the use of certain medications, including selective serotonin reuptake inhibitors; serotonin–norepinephrine reuptake inhibitors, such as tramadol; monoamine oxidase inhibitors; and other medications that increase serotonin levels [44]. Some recreational drugs also increase serotonin levels. Although posttraumatic osteoarthritis has not been directly correlated with serotonin syndrome, tramadol, which is taken for pain management in posttraumatic osteoarthritis, may increase the risk of serotonin syndrome. For example, patients with posttraumatic osteoarthritis who are taking tramadol may be at increased risk of serotonin syndrome if they are also taking selective serotonin reuptake inhibitors or other drugs that increase serotonin levels. Health-care providers must carefully consider the potential risks and benefits of medications, particularly those that increase serotonin levels, when treating patients with posttraumatic osteoarthritis. Patients should be educated about the signs and symptoms of serotonin syndrome—which can include agitation, confusion, a rapid heart rate, high blood pressure, dilated pupils, muscle rigidity, and fever—and be instructed to seek immediate medical attention if they experience these symptoms.

Our study found that male patients had a higher risk of hip fracture than female patients after long-term use of tramadol for pain control in older adults with posttraumatic osteoarthritis. Possible explanations include that men may be more likely to engage in activities and may be prescribed higher doses of tramadol compared to women, which could increase their risk of falls and subsequent hip fractures. Alternatively, men may be less likely to seek medical attention for pain or report falls, which could result in delayed diagnosis and treatment of hip fractures. We also found that patients aged 60–70 years had a higher risk of hip fracture than those aged 70 years or older after long-term use of tramadol for pain control in older adults with posttraumatic osteoarthritis. The possible explanation is that older adults aged 70 years or older may be more likely to have other comorbidities or medical conditions that increase their risk of falls and hip fractures, which could attenuate the effect of tramadol use on hip-fracture risk. Alternatively, older adults aged 70 years or older may be more likely to modify their behavior and take precautions to prevent falls, such as using walking aids or reducing physical activity, which could reduce their risk of hip fractures. On the other hand, older adults aged 60–70 years may be more active and engage in more physically demanding activities that increase their risk of falls and subsequent hip fractures.

This study has several limitations. First, the clinical data, such as patient lifestyle, perception level, bone quality, physical function status, degree of pain of the affected sites, body mass index, and laboratory examination data could not be retrieved from the Taiwan National Health Insurance Research Database. So, we cannot rule out the presence of confounding factors that might be relevant to the risk of hip fracture despite strict propensity score matching and adjustment. Although we aimed to adequately control for potential confounding effects by using propensity score matching and a multivariate Cox proportional hazards model, unknown or unmeasured confounders may still be present in this nonrandomized observational study. Second, although data on the frequency and duration of tramadol prescriptions were available in the Taiwan National Health Insurance Research Database, those for drug compliance were not. Third, we did not put the detailed medications of individuals as the risk factors analyzed in this study. Although we have excluded those who received a regular steroids, sedative, hypnotic, or anticonvulsant prescription (a more-than-90-day within 1 year prescription) and those who were diagnosed as having a mental disorder in our study, there may still be other medications that cause patients to have varying degrees of hip-fracture risk.

Despite these limitations, this study demonstrated the significant effect of long-term tramadol medication for older adults with posttraumatic osteoarthritis for pain control on the increase of new hip fractures requiring surgery. This is the first cohort study to focus on the association between long-term tramadol use and hip fracture among older adults with posttraumatic osteoarthritis, and the power of the association is highly likely to be sufficient because of the representative sample of 2 million people from a nationwide research database and the long follow-up period. For patients at a high risk of hip fracture, those aged more than 60 years (especially those aged 60–70 years), and those who are male, tramadol for long-term continuous use should only be prescribed with caution. Personalized and customized pain control for every patient with posttraumatic osteoarthritis is our goal for providing the most benefit and minimizing the adverse effects of medical treatment. For patients who have a high risk of serotonin syndrome, respiratory inhibition, or hip fracture but who still need to adequately control their pain from posttraumatic osteoarthritis, physicians may adjust their dose and frequency of tramadol or combine it with other kinds of analgesics. If the patients with a high risk of adverse effects still need tramadol as effective pain control medication, physicians or nurses should inform them frequently of the relative risks and adjust medication doses based on their health conditions during their hospital visits. Future interventional and comparative designs of research may apply to evaluate the most personalized dosages of tramadol for non-cancerous major limb joint pain for every individual patient, so that the patient can get best relief of pain and the least side effects from effective analgesics such as tramadol.

## 5. Conclusions

Tramadol as a long-term (at least 90 continuous days within one year) pain control analgesia for older adults with posttraumatic osteoarthritis appears to increase the risk of hip fracture, especially among patients aged 60–70 years and those who are male. Although tramadol is a commonly used pain medication, its possible adverse effects among patients at a high risk of hip fracture should be taken into consideration when prescribing it for pain control. Health-care providers must weigh the potential risks and benefits of tramadol use in patients with osteoarthritis or other conditions that cause joint pain and monitor patients for adverse effects, including falls and fractures. Patients taking tramadol should be advised to take precautions to prevent falls, such as using assistive devices and ensuring adequate lighting in their homes. Our study highlights the potential risks associated with long-term use of tramadol for pain control in older adults with posttraumatic osteoarthritis, and healthcare professionals should consider the risks and benefits when prescribing tramadol to this population.

## Figures and Tables

**Figure 1 jpm-13-00580-f001:**
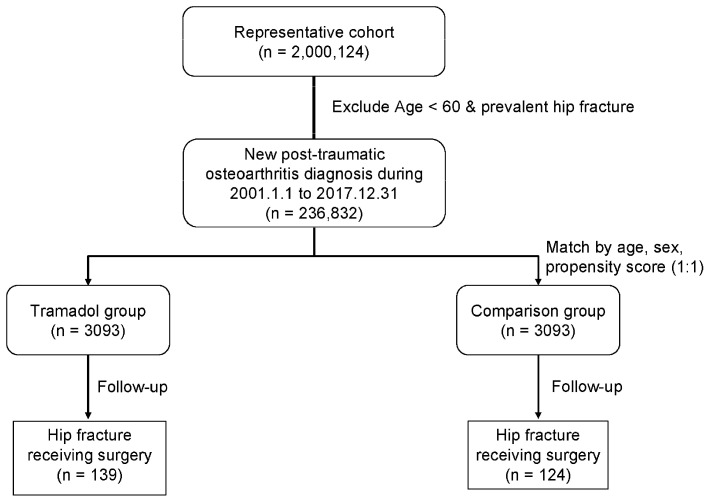
Study design flowchart.

**Figure 2 jpm-13-00580-f002:**
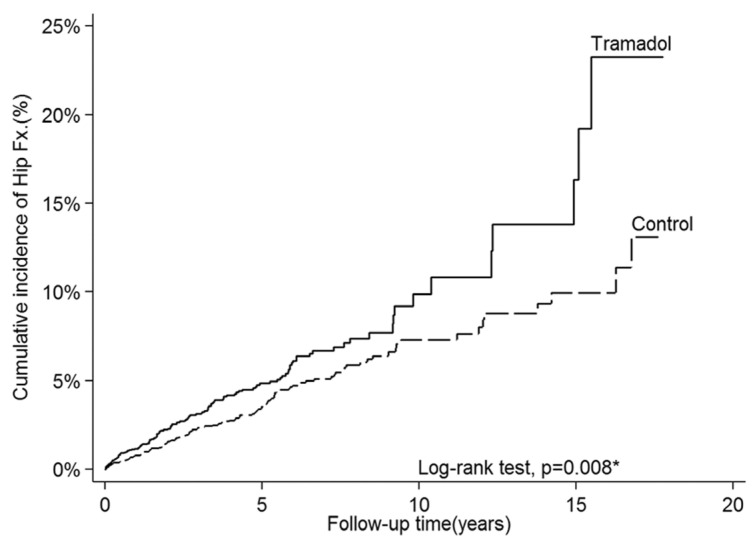
Cumulative hip fracture incidence was estimated using the Kaplan–Meier method, and a log-rank test was used to compare the incidence curves. * *p* < 0.05 was considered statistically significant.

**Table 1 jpm-13-00580-t001:** Patient demographic data.

Variables	Control	Tramadol	Total	*p*-Value
N	3093	3093	6186	
Age	70.04 ± 7.63	70.04 ± 7.62	70.04 ± 7.63	0.987
Age group	-	-	-	0.998
60–70 y/o	1658 (53.6%)	1657 (53.6%)	3315 (53.6%)	
70–80 y/o	1034 (33.4%)	1036 (33.5%)	2070 (33.5%)	
≥80 y/o	401 (13.0%)	400 (12.9%)	801 (13.0%)	
Gender	-	-	-	0.680
Female	1297 (41.9%)	1313 (42.5%)	2610 (42.2%)	
Male	1796 (58.1%)	1780 (57.6%)	3576 (57.8%)	
Hypertension (%)	1350 (43.6%)	1339 (43.3%)	2689 (43.5%)	0.778
Diabetes (%)	694 (22.4%)	695 (22.5%)	1389 (22.5%)	0.976
Hyperlipidemia (%)	594 (19.2%)	600 (19.4%)	1194 (19.3%)	0.847
Coronary artery disease (%)	380 (12.3%)	387 (12.5%)	767 (12.4%)	0.787
Cerebral vascular accident (%)	248 (8.0%)	269 (8.7%)	517 (8.4%)	0.335
Chronic liver disease (%)	175 (5.7%)	171 (5.5%)	346 (5.6%)	0.825
Chronic renal failure (%)	160 (5.2%)	159 (5.1%)	319 (5.2%)	0.954
Depression (%)	114 (3.7%)	112 (3.6%)	226 (3.7%)	0.892
Osteoporosis (%)	96 (3.1%)	109 (3.5%)	205 (3.3%)	0.356
Follow-up Time (yr.)	5.17 ± 3.99	3.79 ± 3.27	4.48 ± 3.71	

Data are presented as n or mean ± standard deviation.

**Table 2 jpm-13-00580-t002:** Risk of hip fracture in patients with versus without tramadol use.

Variables	Tramadol	
	Yes	No
Patient numbers	3093	3093
Hip fx. cases	139	124
Person-years	11,712	15,997
Incidence rate ^a^	11.9	7.8
Univariate model		
crude HR (95% CI)	1.41 (1.09–1.82)	1 (ref.)
*p*-value	0.008 *	
Multivariate model ^b^		
aHR (95% CI)	1.41 (1.09–1.82)	1 (ref.)
*p*-value	0.008 *	

HR, hazard ratio; aHR, adjusted hazard ratio; CI, confidence interval; and ref., reference. ^a^ Per 1000 person-years. ^b^ Multivariate Cox proportional hazard regression model with adjustment for all baseline characteristics. * *p* < 0.05 was considered statistically significant.

**Table 3 jpm-13-00580-t003:** Subgroup analysis of Cox’s regression model for the association between tramadol use and risk of hip fracture by age and gender.

Variables	Crude HR (95% CI)	*p*-Value	Adjusted HR (95% CI)	*p*-Value
Main model				
No	1.00		1.00	
Yes	1.41 (1.09–1.82)	0.008 *	1.41 (1.09–1.82)	0.008 *
Age				
60–70 y/o				
No	1.00		1.00	
Yes	2.17 (1.32–3.57)	0.002 *	2.11 (1.29–3.47)	0.003 *
70–80 y/o				
No	1.00		1.00	
Yes	1.13 (0.77–1.64)	0.537	1.09 (0.75–1.59)	0.656
≥80 y/o				
No	1.00		1.00	
Yes	1.38 (0.84–2.25)	0.203	1.35 (0.82–2.20)	0.233
Gender				
Male				
No	1.00		1.00	
Yes	1.75 (1.19–2.58)	0.005 *	1.83 (1.24–2.70)	0.002 *
Female				
No	1.00		1.00	
Yes	1.15 (0.82–1.62)	0.407	1.16 (0.83–1.63)	0.391

HR, hazard ratio; CI, confidence interval. * *p* < 0.05 was considered statistically significant.

## Data Availability

All data generated or analyzed during this study are included in this published article.

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
