# Peer review of "Tramadol May Increase Risk of Hip Fracture in Older Adults with Post-Traumatic Osteoarthritis"

_jpm, 2023, doi:10.3390/jpm13040580_

Round 1
Reviewer 1 Report
This study investigated the association between long-term use of tramadol for pain control and subsequent hip fractures in patients aged 60 years or older with posttraumatic osteoarthritis. It was found that Tramadol use was identified as a risk factor for hip fracture, especially among patients aged 60–70 years and among male patients. This is a population-based retrospective cohort study. A multivariate Cox proportional hazards model was used to calculate the hazard ratios. Generally, I have a good impression with this manuscript. But still there are some concerns listed below.
1. Which part of the traumatic arthropathy cases were included is important information in discussing the risk. It should be described in the methods and results.
2. Do the authors have any information about the degree of pain for each group?
3. How were the uses of other medications other than analgesics?
4. As the authors described in the limitation, the lifestyle of patients may also affect the risk of hip fractures. The Tramadol group may have experienced more pain than the control group. In that case, the higher incidence of hip fracture in Tramadol group may be caused by their lifestyle or moving ability. The authors need to justify these factors.
5. Legends for Table 3 should be corrected.
Author Response
Dear Reviewer:
Thank you very much for all of your reminding and valuable suggestions.
This study investigated the association between long-term use of tramadol for pain control and subsequent hip fractures in patients aged 60 years or older with posttraumatic osteoarthritis. It was found that Tramadol use was identified as a risk factor for hip fracture, especially among patients aged 60–70 years and among male patients. This is a population-based retrospective cohort study. A multivariate Cox proportional hazards model was used to calculate the hazard ratios. Generally, I have a good impression with this manuscript. But still there are some concerns listed below.
- Which part of the traumatic arthropathy cases were included is important information in discussing the risk. It should be described in the methods and results.
Ans: Thank you for your reminding. We have clarified our inclusion as “ the patients with a diagnosis of posttraumatic osteoarthritis of any major joint of upper limbs or lower limbs after trauma” in method and result sections
- Do the authors have any information about the degree of pain for each group?
Ans: No specific degree of pain of posttraumatic osteoarthritis of every patient was indeed a limitation of this population-based cohort study. For more accuracy representation of patients’ conditions indicated for analgesics prescriptions, we included the patients with a diagnosis of posttraumatic osteoarthritis for at least 3 times of outpatient department treatment or at least 1-time inpatient admission treatment, and a more-than-90-day within 1 year prescription for tramadol and the combinative drugs including tramadol for pain control from 2001 to 2017 were enrolled as the tramadol cohort. The patients who did not use tramadol but used any non-steroidal anti-inflammatory drugs, paracetamol, or muscle relaxants were matched with the patients in the tramadol cohort. Our prescription of the long-term analgesics must meet the criteria of National Health Insurance as the patients should have moderate to severe pain from the diagnosis. So, the included patients in this study must have moderate to severe pain form posttraumatic osteoarthritis to meet the design criteria of this study. Still, we put your suggestion into one of our limitations.
- How were the uses of other medications other than analgesics?
Ans: In this study we did not record all detailed medications of individual patient as one of our research parameter. But we have excluded some medication with the major concern of fall down or hip fracture as exclusion criteria. We have made the supplement description into our exclusion criteria as below: “ (4) those who received regular steroids, sedative, hypnotic, or anticonvulsants prescription (more-than-90-day within 1 year prescription) (5) those who were diagnosed as mental disorder.” We also put your suggestion into one of our limitations.
- As the authors described in the limitation, the lifestyle of patients may also affect the risk of hip fractures. The Tramadol group may have experienced more pain than the control group. In that case, the higher incidence of hip fracture in Tramadol group may be caused by their lifestyle or moving ability. The authors need to justify these factors.
Ans: Thank you for your suggestion. This indeed was major limitations of the population-based study and listed in our limitation section. We have made some efforts to precise our analytic results as below: 1. Strict inclusion/exclusion criteria : The patients with a diagnosis of posttraumatic osteoarthritis for at least 3 times of outpatient department treatment or at least 1-time inpatient admission treatment, and a more-than-90-day within 1 year prescription for tramadol and the combinative drugs including tramadol for pain control from 2001 to 2017 were enrolled as the tramadol cohort. The index date was the first date of prescription. The exclusion criteria were (1) tramadol prescription claims in 2000 (to ensure that patients were new users); (2) those who were younger than 60 years; and (3) the diagnosis of a hip fracture before the index date (4) those who received regular steroids, sedative, hypnotic, or anticonvulsants prescription (more-than-90-day within 1 year prescription) (5) those who were diagnosed as mental disorder. The patients who did not use tramadol but used any non-steroidal anti-inflammatory drugs, paracetamol, or muscle relaxants were matched with the patients in the tramadol cohort; 2. Propensity score matching at a 1:1 ratio was performed for age, sex, index year, and the comorbidities for both comparison groups. Still, there were still some missing data crucial to the results of this study. But we believe the results of this study still point out the important notice of correlation between hip fracture risk and long-term tramadol use. Using tramadol for too long period may make the older adults in pain dependent on medications, or not want to solve the pain problem in other positive ways, which may cause consequences as severe as hip fractures.
- Legends for Table 3 should be corrected.
Ans: Thank you for your reminding. We have modified the legend to be as below description: “Subgroup analysis of Cox's regression model for the association between tramadol use and risk of hip fracture by age and gender”

Reviewer 2 Report
(1) I think that the introduction could mention clearly the background, related studies, and objective.
(2) Is there a reference for the Taiwan National Health Insurance Research Database? Is this a public dataset?
(3) I think that the procedure of data correction is suitable.
(4) Continuous variables were compared using a parametric test (Student’s t-test). Did you confirm data distributions of these variables before the Student's t-test?
(5)I think that resolution of Fig.2 should be improved.
(6)I think that the temporal change of cumulative hip fracture incidence in the Tramadol group is interesting (Fig.2). This graph showed that cumulative incidence suddenly increased around 15 years later. If possible, please write a discussion about this temporal change.
(7) This study provided new knowledge about the relationship between the risk of hip fracture and using Tramadol. In addition, several recommendations for to use of Tramadol in the long term will contribute to the treatment of Post-Traumatic Osteoarthritis.
(8) How will you investigate appropriate analgesic dosages? Will future researches require intervention?
Author Response
Dear Reviewer:
Thank you very much for all of your reminding and valuable suggestions.
Response to Reviewer 2
- I think that the introduction could mention clearly the background, related studies, and objective.
Ans: Thank you for your encouragement.
(2) Is there a reference for the Taiwan National Health Insurance Research Database? Is this a public dataset?
Ans: Thanks for your kind reminding. There was a paper which provided an overview of Taiwan’s National Health Insurance Research Database (NHIRD) [1]. The database was currently administrated by Taiwan’s Ministry of Health and Welfare (MOHW). The government established a Health and Welfare Data Center (HWDC), a data repository site that centralizes the NHIRD and about 70 other health-related databases for data management and analyses. Researchers could apply for them after their research plans were approved by Institutional Review Board. We also have added this reference into our Material and Method section.
<Reference>
- Hsieh CY, Su CC, Shao SC, Sung SF, Lin SJ, Kao Yang YH, Lai EC. Taiwan's National Health Insurance Research Database: past and future. Clin Epidemiol. 2019 May 3;11:349-358. doi: 10.2147/CLEP.S196293. PMID: 31118821; PMCID: PMC6509937.
(3) I think that the procedure of data correction is suitable.
Ans: Thank you for your encouragement.
(4) Continuous variables were compared using a parametric test (Student’s t-test). Did you confirm data distributions of these variables before the Student's t-test?
Reply: Thanks for your kind reminding. We had confirmed that the data of continuous variables followed normal distribution via Kolmogorov-Smirnov test (K-S test).
(5) I think that resolution of Fig.2 should be improved.
Ans: Thanks for your suggestion. We had improved the resolution of Figure 2 to 300 dpi. Please refer to the updated files.
(6) I think that the temporal change of cumulative hip fracture incidence in the Tramadol group is interesting (Fig.2). This graph showed that cumulative incidence suddenly increased around 15 years later. If possible, please write a discussion about this temporal change.
Ans: Thanks for providing your great insight. We thought there could be two possible reasons. Firstly, the incidence of hip fracture increases with age according to previous study [1]. As the figure shown below, the incidence of hip fracture increases dramatically after 80 y/o. Since the mean age of our cohort was around 70 y/o, the incidence of hip fracture increased dramatically after 15 years later in both groups. Secondly, the number of patients remained in cohort might decrease as the follow-up time increased. Thus, the incidence (number of incident cases/number of people at risk) might increase dramatically as the denominator decreased. We have added the related content into the first paragraph of Discussion.
<Reference>
- Chie WC, Yang RS, Liu JP, Tsai KS. High incidence rate of hip fracture in Taiwan: estimated from a nationwide health insurance database. Osteoporos Int. 2004 Dec;15(12):998-1002. doi: 10.1007/s00198-004-1651-0. Epub 2004 May 20. PMID: 15156304.
(7) This study provided new knowledge about the relationship between the risk of hip fracture and using Tramadol. In addition, several recommendations for to use of Tramadol in the long term will contribute to the treatment of Post-Traumatic Osteoarthritis.
Ans: Thank you for your suggestion. I have added the following recommendation to the last paragraph of Discussion: “For patients who have a high risk of serotonin syndrome, respiratory inhibition, or hip fracture but who still need to adequately control their pain from posttraumatic osteo-arthritis, physicians may adjust their dose and frequency of tramadol or combine it with other kinds of analgesics. If the patients with high risk of adverse effects still need tramadol as effective pain control medication, physicians or nurses should inform the relative risks to them frequently and adjust medication doses based on their health conditions during their hospital visits. “
(8) How will you investigate appropriate analgesic dosages? Will future research require intervention?
Ans: Thank you for your suggestion. I have added the following investigative idea to the last paragraph of Discussion: “Future interventional and comparative designs of research may apply to evaluate the most personalized dosages of tramadol for non-cancerous major limb joint pain for every individual patient, so that the patient can get best relief of pain and least side effect from the effective analgesics such as tramadol.
